# Evaluation of the Influence of Intervention Tools Used in Nutrition Education Programs: A Mixed Approach

**DOI:** 10.3390/nu17152460

**Published:** 2025-07-28

**Authors:** Luca Muzzioli, Costanza Gimbo, Maria Pintavalle, Silvia Migliaccio, Lorenzo M. Donini

**Affiliations:** Department of Experimental Medicine, Sapienza University, 00185 Rome, Italy; gimbo.1888115@studenti.uniroma1.it (C.G.); maria.pintavalle@uniroma1.it (M.P.); silvia.migliaccio@uniroma1.it (S.M.); lorenzomaria.donini@uniroma1.it (L.M.D.)

**Keywords:** nutrition education, adherence to Mediterranean diet, KIDMED, healthy diets, schoolchildren

## Abstract

**Background**: In a global panorama marked by a progressive rise in obesity, metabolic syndrome, and chronic non-communicable disease prevalence, nutrition education (NE) might play a pivotal role in restoring adoption and strengthening adherence to dietary patterns that protect human health. Therefore, the primary purpose of this work is to review the existing scientific literature studying NE programs aimed at schoolchildren in the decade 2014–2024 and evaluate the effectiveness of intervention tools. **Methods**: During the first phase of this research, a qualitative analysis was conducted to track similarity in intervention tools and strategies used in nutrition education programs. In the second phase, a quantitative analysis was carried out, extracting common parameters among studies and assessing their potential influence in improving adherence to the Mediterranean diet (MD). **Results**: A high degree of heterogeneity was observed in educational program designs and intervention tools, which were usually not properly described and justified. All studies that measured adherence to the MD registered an improvement after the intervention, in some cases even higher than 10%. However, this study found no relationship between common parameters (i.e., number of formal tools, number of non-formal tools, lesson duration, and program length) used in NE and the improvement in students’ adherence to MD. **Conclusions**: This research has contributed to outlining a general framework of NE and to promoting a systematic approach in this research field.

## 1. Introduction

Due to the industrialization and globalization of food systems, modern dietary habits are characterized by a high consumption of animal products and a large amount of so-called empty calories—provided by energy-rich, nutrient-poor processed foods—leading to high worldwide rates of overweight and obesity [1,2,3]. These modern food consumption patterns are associated with metabolic non-communicable diseases that are growing at epidemic levels [4]. The so-called “pandemic” of obesity is, therefore, one of the consequences of the sudden changes in nutritional regimes that have taken place over the last few decades [5]. Obesogenic environments influence eating behaviors, discouraging access to fresh, locally grown fruits and vegetables [6]. The progressive move away from a diet based on local, unprocessed products—which began in the second half of the 20th century—has led to an increased caloric intake with an excessive consumption of simple sugars, sugar-sweetened beverages, and animal-derived and ultra-processed foods at the expense of plant-based options [7]. The consequence of this is that fruit and vegetable intakes are now under the recommended levels, especially within the child population [8,9,10].

In Italy, 25.9% of children do not consume fruit and/or vegetables daily, and 37% of them eat legumes less than once a week [11]. The 2022 WHO Europe report states that 59% of European adults and almost 1 in 3 children are overweight or obese [12]. Despite prevention and treatment efforts, obesity currently remains an international health problem: according to the World Health Organization, childhood overweight and obesity have become one of the most serious public health concerns [13,14,15,16,17]. Thus, it is not surprising that obesity and poor adherence to healthy diets—such as the Mediterranean diet (MD)—have turned out to be related as this reflects the changes in eating habits, which are increasingly shifted towards more “Western” diets [18,19]. Moreover, growing up in urbanized and obesogenic environments, children have been disconnected from the agricultural provenance of food products, which makes them increasingly exposed to the risk of developing obesity-related disorders [8]. Conversely, there is a significant association between adherence to a healthy eating model and the literacy of the population in terms of better nutritional knowledge [20]. This fact confirms the urgent need to intervene with nutritional education, leveraging local eating patterns that encourage healthy and sustainable habits, such as the MD in the Mediterranean area [20]. The early introduction of nutrition education (NE) programs is crucial for maintaining healthy behaviors through adulthood [14,21,22,23]. Indeed, improving diet quality and nutritional knowledge appears to be one of the main strategies to prevent childhood obesity, together with shaping environments and communities in a way that aids everyone in making informed choices easily, at any time [6]. To this end, NE proves to be an essential key factor.

Thus, the aim of this study is to assess which are the most effective intervention tools, through NE programs, to increase adherence to healthy eating patterns in children and adolescents.

## 2. Materials and Methods

This study is designed as mixed-methods research that uses a sequential exploratory strategy. It involves a “first phase of qualitative data collection and analysis followed by a second phase of quantitative data collection and analysis that builds on the results of the first qualitative phase” [24]. The mixed approach combines a literature review with a quantitative analysis that is used to evaluate the influence of NE tools on intervention improvement, measured by adherence to healthy diets.

### 2.1. Search Strategy

The bibliographic databases that were searched included PubMed, ResearchGate, and Scopus. The search was conducted in December 2024. The relevant literature was identified using the following keywords, according to the different possible combinations: (nutrition education intervention) AND (children) AND (Mediterranea* OR “healthy and sustainable diets”). 

Since in most studies the evaluation of efficacy was carried out using the KIDMED questionnaire (to measure the differences pre- and post-intervention), for the quantitative analysis stage, it was decided to deepen the research by including the following search terms: “KIDMED” AND “nutrition education”; “KIDMED questionnaire” AND “education”.

### 2.2. Eligibility Criteria

The eligibility criteria for the inclusion of the publications were as follows:Peer-reviewed studies involving human populations under the age of 18 years old;Experimental studies that investigated a nutrition education intervention;Studies that assessed, pre- and post-intervention, the population’s adherence to healthy diets;Studies published between January 2014 and November 2024.

No study was excluded based on geographical origin to maintain a broad vision on the subject. Studies were excluded in cases where the nutritional education intervention concerned specific populations or groups of hospital patients, or where the improvement was related to a single aspect of nutrition (e.g., patients with coeliac disease). Abstracts of complete and available publications were assessed for suitability in accordance with the inclusion criteria.

### 2.3. Qualitative Analysis Phase: Data Collection and Extraction

The characteristics of the articles considered to meet the inclusion criteria were collected on an Excel spreadsheet. For each article, the following information was extracted using a standardized format: First Author (year of publication), Objectives, Population, Intervention Methods, Lesson Duration/Program Length, Outcome, Questionnaire, and Results.

Intervention tools were classified into two categories: formal and non-formal tools. Formal tools refer to educational strategies that are included in a structured and tiered education pathway that maintains a front-of-class approach. This category includes oral presentations, textbooks, webinars, videos, Q&A sessions, group discussions, and reflections [25]. Non-formal educational tools, e.g., learning by participation and peer education, focus primarily on intellectual, emotional, social, and behavioral aspects rather than on cognitive performance as in formal tools. The learning of the contents happens through activities that stimulate all five senses and artistic expression, as well as by means of games and physical activity, thus overcoming the exclusive and unilateral interaction between student and teacher [26].

### 2.4. Quantitative Analysis Phase: Data Collection, Extraction, and Coding

For the quantitative analysis, the literature did not always provide unbiased evidence of the effectiveness of the NE program implemented, and the methods of measurement were often different; therefore, in order to be able to make comparisons, it was decided to select homogeneous studies, restricting them to those that used the Mediterranean Diet Quality Index for Children and Adolescents questionnaire (KIDMED). In fact, the KIDMED questionnaire is the most frequently used in the literature on NE in the area of greatest interest for this study: the Mediterranean region. Data were extracted and reported into an Excel table as follows: “First Author (year of publication)”, “Population”, “Intervention Methods”, “Lesson Duration/Program Length”, “Results (Δ KIDMED)”, and “KIDMED Improvement (%)”. To perform the quantitative analysis, the data were further organized into five variables that were identified as common parameters among all selected studies: the number of “Formal Intervention Strategies” (e.g., frontal teaching tools); the number of “Non-formal Intervention Strategies” (e.g., active education tools); “Lesson Duration” (expressed in minutes); “Program Length” (expressed in months); and “KIDMED Improvement” (expressed as a percentage).

### 2.5. KIDMED Questionnaire

Studies included in the quantitative analysis assessed the effectiveness of the interventions using the KIDMED questionnaire (Table 1). This questionnaire was developed and validated in 2004 by a group of scholars led by Serra-Majem [27]. As reported by the authors, the KIDMED “*index ranges from 0 to 12 and is derived from a 16-item. Items indicating negative aspects of the Mediterranean dietary pattern are scored –1, while those reflecting positive adherence are scored +1. Based on the total score, adherence is classified into three categories: (1) >8, indicating optimal adherence to the Mediterranean Diet; (2) 4–7, suggesting a need for dietary improvement; and (3) ≤3, reflecting very poor dietary quality*” [27].

### 2.6. Evaluation of the Effectiveness

The effectiveness of educational interventions was considered as the improvement in the KIDMED value, calculated as the percentage ratio between the incremental value of the post-intervention KIDMED score (named Δ KIDMED) and the maximum score of the KIDMED test (i.e., 12).KIDMED Improvement (%) = (KIDMEDpost − KIDMEDpre)/KIDMEDmax × 100(1)

Then, several hypotheses were formulated to quantitatively analyze possible relationships between program effectiveness and the following parameters extracted from the selected studies (Figure 1):Number of educational tools divided by category: formal (e.g., frontal teaching) or non-formal tools (e.g., active education tools);Duration of individual meetings;Total length of the educational intervention.

### 2.7. Statistical Analysis

Statistical analysis was carried out with SPSS (IBM SPSS Statistics for Mac, Version 28.0. Armonk, NY, USA: IBM Corp.). The Shapiro–Wilk test was performed on variables extracted from the selected studies, suggesting a non-normal distribution for most of them (i.e., number of forsmal tools, number of non-formal tools, and educational program length). Therefore, the non-parametric test of Kendall’s tau-b was applied to assess the correlation between the variables of interest. Due to the limited sample size (N = 11), Kendall’s tau-b test was preferred to Spearman’s rho ranking test for its higher robustness to outliers, the narrower confidence intervals it produces, and because it performs well with small sample sizes [28,29]. A principal component analysis was then used for dimensionality reduction before applying a multiple linear regression to investigate possible relationships between the dependent variable (i.e., the improvement in MD adherence) and the four common parameters extracted from educational programs (independent variables) (Figure 1) [30]. A forward linear regression was further applied to build models by iteratively adding the most statistically significant predictor variable. Lastly, a hierarchical cluster analysis was performed to further identify possible patterns and relationships within studies; a dendrogram was used to graphically display the obtained clusters.

## 3. Results

### 3.1. Study Selection

The database search identified a total of 110 articles whose suitability was assessed through the analysis of the abstracts to arrive at a total number of 45 articles considered as meeting the inclusion criteria for the qualitative analysis (Figure 2). Given the need to evaluate the effectiveness of educational interventions, 10 other articles were excluded because they did not allow for an evaluation of the intervention tools adopted. To compare the various intervention tools and carry out a quantitative analysis of effectiveness, a final pool of 11 articles reporting the use of the KIDMED questionnaire was included.

### 3.2. Qualitative Analysis

#### 3.2.1. Population

The qualitative analysis of the articles has shown that most of the educational interventions were implemented in the school context (29 out of 45 studies), mostly primary schools. In fact, despite the age of the reference population varying from 18 months to 19 years old, the majority of studies targeted young students aged between 5 and 12 (n = 28), followed by studies on adolescents (n = 9). In nine cases, the involvement of adults took place exclusively as parents or caregivers, whereas three studies also included teachers in the target population. A summary of all included studies is available in the Appendix A.

#### 3.2.2. Geographical Areas

The geographical origin of the studies referred to different continents (Figure 3). A high prevalence was recorded in the Mediterranean area, with nine Spanish, six Italian, two Portuguese, and two Greek studies, respectively. Other studies implemented in Europe include Belgium, Sweden, Poland, Croatia, and Germany. With regard to other continents, six studies were registered in Africa (three of which were in Ghana), five in North America (US), and two in Australia. Each of the following countries has also been found to have conducted one study: Japan, Lebanon, Indonesia, Tunisia, Cyprus, Malaysia, Iraq, The Philippines, Egypt, China, Taiwan, Brazil, Chile, Iran, and Nepal.

#### 3.2.3. Types of Interventions

Most educational interventions included both formal and non-formal tools. Teachers, as well as parents, were often involved in the educational activities: in some cases, their participation was limited to receiving newsletters [14,31] or a one-day nutrition education training [32]; in other cases, it consisted of multiple nutrition education sessions [33,34,35,36,37] or other activities such as cooking classes [31,34,38,39]. In eight studies, schoolteachers themselves implemented the educational program; in other cases, meetings were conducted by external professionals, often dieticians or nutrition experts. In six studies, the educational program was accompanied by an intervention of optimization or modification of the school lunch menu, usually aimed at reducing food waste and the environmental impacts of school meals.

The topics addressed within the school context varied according to the specific objective of the study, ranging from hygiene, daily portions, and physical activity to food groups, reaching in some cases the theme of food waste and sustainability. The most recurrent topics, at the core of the educational programs, were related to healthy eating guidelines, healthy eating patterns and habits, daily servings, and the importance of fruits, vegetables, and physical activity [8,39,40,41,42,43,44,45,46,47].

#### 3.2.4. Educational Strategies and Intervention Tools

The literature search has shown that the strategies (duration, timing, and use of questionnaires) and intervention tools used during educational programs varied greatly from case to case. Pre- and post-intervention assessments of diet adherence were of different kinds, such as standardized or tailor-made questionnaires based on the healthy eating guidelines of the country in which the intervention took place.

Due to the high heterogeneity found in the intervention methods, they were first divided into formal and non-formal tools. The results show a preponderance towards the use of instruments of active learning and participatory education. In fact, of the 45 articles found suitable, 35 implement an NE program using both front-end and participatory tools, while only 10 choose to adopt exclusively front-end teaching tools.

Formal tools were employed to present and/or discuss the topic of the lectures. In fact, the most frequently used tools were group discussion activities (n = 17) and presentations (n = 16), followed by booklets/leaflets, flyers, brochures, and the infographic of the Food Pyramid. The MyPlate infographic was also adopted in six studies, along with textbooks, which were different each time depending on the country of implementation and the program’s topic. Non-formal instruments and strategies were numerous, scattered, and usually not justified. The use of video was the most adopted tool, followed by the creation, preparation, or reformulation of healthy recipes or snacks, as well as exercise- and activity-oriented games. Board games (such as the Game of the Goose and Memory) [16,37,48,49], group games, and team competitions [33,34,35] were also frequently adopted, while interpretations of plays, performances, and role-playing have been mentioned several times [3,14,17,19,50,51], followed by visits to local farms [1,8], educational gardens [1,8,31,38,39,45], the use of quizzes [18,39], the creation of posters [1,8,14,17,40,41,44,50,52,53,54], the development of artistic expression artefacts [14,23,33,50,53,55], and the organization of workshops [31,37,39,56,57,58,59].

#### 3.2.5. Intervention Lengths and Single-Meeting Durations

Another variable confirming the substantial diversity among the studies considered is the heterogeneous nature of the intervention durations, both in terms of the total extent of the investigations and of the length of the individual meetings [37]. It was very difficult to conduct a qualitative and quantitative assessment of the duration of programs and their impact on the effectiveness of interventions. Not all articles specified the duration of the individual meetings and the total length of the investigation. Also, the frequency of meetings was only seldom specified (e.g., “once a week” or “once a month”). Overall, the single sessions ranged from 15 min to 2 consecutive hours, most of which lasted between 40 and 90 min. Instead, the length of the intervention varied from 3 weeks to 3 years, with “3 months” as the modal value.

### 3.3. Evaluation and Quantification of Effectiveness

Only 37 studies assessed the impact of their intervention on the improvement in eating habits of the population surveyed. Among them, 21 studies quantified it. Eleven studies measured the improvement in adherence to the Mediterranean diet, whereas seven only evaluated the increase in fruit and vegetable consumption. Another nine research groups measured the improvement in subject knowledge in terms of nutritional knowledge. Only one justified its choices in terms of the methodology used [58].

In view of this lack of homogeneity, only those studies (n = 11) that measured adherence to the MD using the KIDMED questionnaire were taken into account in order to quantify the effectiveness of interventions and make comparisons (Table 2). To perform this, a series of quantitative variables were extracted by the identification of common parameters reported in studies, i.e., number of formal tools, number of non-formal tools, lesson duration, and length of educational interventions.

All studies reported an improvement in adherence to the MD. Ameliorations ranged from 0.4% to 15.77%, with four studies observing low (≤0.50 Δ KIDMED score) [19,36,52,60], four reporting medium (from 0.51 to 1.00) [37,58,61,62], and three indicating high (>1.00) improvements [33,44,57]. Studies with the highest effectiveness adopted a multi-component approach, mixing formal and non-formal education tools; lesson durations varied from 15 to 120 min, whereas interventions lasted from 6 weeks to 3 years. When Kendall’s tau-b test was performed to assess possible correlations between variables of interest, no significant relationships emerged either within common parameters or with KIDMED improvement (Table 3).

Moreover, principal component analysis indicated that variance in common parameters was sufficiently high to insert all four variables as potential predictors into the multivariate linear regression test. Therefore, a multiple linear regression was calculated predicting educational program capacity to improve adherence to the MD based on whether they included formal training, the number of formal and non-formal activities, the duration of the class sessions, and the length of each educational program. The regression equation was not significant (F(4, 6) = 0.416, *p* > 0.05) with an R^2^ of 0.217. None of the independent variables were found to be a significant predictor of diet adherence improvement. In addition, none of the models built in the forward regression analysis demonstrated a statistical significance by progressively adding the most influential predictor, thus confirming the null hypothesis.

Lastly, a hierarchical cluster analysis was performed to group similar studies together based on the four common parameters. The dendrogram is depicted in Figure 3, where studies were divided into high (green), medium (amber), and small (red) KIDMED improvements. Except for studies by Roset-Salla [37], Roccaldo [62], and Mahmood [58]—which shared a lesson duration of 90’ and a medium KIDMED improvement—the other sub-clusters did not show any common parameters in combination with a similar effectiveness in changing adherence to the Mediterranean diet.

## 4. Discussion

A high degree of heterogeneity was observed in the educational program designs and intervention tools, which were usually not properly described and justified. When focusing on studies related to NE programs targeted to increase adherence to the MD, all interventions registered an improvement in the KIDMED score. However, Kendal’s correlation and multiple linear regression tests showed a lack of relationship between the number of formal tools, the number of non-formal tools, seminar and program durations, and the adherence improvement.

Several systematic reviews have previously assessed the effectiveness of school-based nutrition programs. They compare nutrition education with other types of interventions, such as changes to food offerings in cafeterias and vending machines, nutrition-friendly school initiatives, or multi-component interventions [63,64,65]. Nutrition education was found to be effective in decreasing body mass index z-score while improving the intake of fruit and vegetables [64], or in modifying diet-related risk factors for obese children and adolescents [66]. All these reviews were centered on educational programs as a whole, without evaluating tools and strategies encompassed in NE interventions [63,64,65]. On the contrary, the present study focuses on the analysis of single nutrition education tools. The research findings indicate that interventions with a multisectoral approach had positive outcomes for children and adolescents. Equally important is the combination of different intervention tools. Multi-component interventions that adopt both lectures/oral presentations and interactive activities/game sessions have a positive impact on children’s nutritional and physical activity behaviors [16]. The most effective strategies often include experiential learning and cross-curricular activities [13]. It appears relevant to use NE tools and strategies that capture the attention of the students, such as different kinds of group activities and games, especially board games [60,67]. Game-based learning is well known for the capacity to stimulate creativity, sociability, and cognitive development, and it can be an effective tool for adopting healthy behaviors [16,48,68].

School garden and interactive game-based nutrition education programs were found to be highly successful in creating active participation in holistic nutrition interventions [7,18]. In this context, parents and teachers seem to exert a strong influence over children’s eating behaviors. It has been found that the increase in children’s food-related knowledge is greater when their guardians’ knowledge increases as well [35,69]. Therefore, the adoption of intervention strategies that include families and school personnel can be crucial [43,70,71]. In addition, since most students have at least one school meal a day, the food offered at school should be part of comprehensive NE programs, following a “whole-school” approach that emphasizes the importance of a healthy diet through a holistic interventional framework [22,24,34,64,69,72].

With respect to the length of interventions and individual meeting durations, a wide variability was found. In both cases, the choice of parameter setting was usually not explained at the methodological level. The systematic review conducted by Murimi et al., 2018, observed that successful programs adopted a multi-component approach and an intervention duration of at least 6 months [73]. In contrast, the present investigation found inconclusive results. In fact, whereas the study by Fernández-Álvarez et al., 2020, suggests that it is preferable to have high-intensity programs lasting at least five or six months, with appointments scheduled at least once a month [52], Roccaldo et al., 2024, evidenced mixed results: both long- and short-duration interventions could achieve successful modifications in children’s dietary habits [62]. Furthermore, D’Adamo et al., 2015, suggest that, to promote dietary habit changes and attitudes toward healthy eating, more than 3 weeks of 1-h weekly sessions are needed with nutrition education interventions that combine different kinds of activities in the school setting [74].

Moreover, the present research did not find any influence on the improvement in diet adherence not only from the intervention length but even from all the other predictor variables, such as educational tools and seminar duration. The correlation analysis observed no significant relationship between the variables considered, and the cluster analysis revealed a lack of association between program parameters and similar effectiveness in terms of adherence to the MD. A possible explanation has been proposed by Fernández-Álvarez et al., 2020, suggesting that the lack of improvement in the KIDMED score that his study obtained (Δ KIDMED_score_ = 0.05) may be attributed to a “ceiling effect” [52]. According to them, the baseline KIDMED score (6.34) was already within an acceptable range and higher than what had been reported in previous studies on comparable populations. However, such adherence is considered moderate [27], and, therefore, far from the 12-point maximum value. In addition, the present study observed results in contrast with that statement: six studies reported a similar baseline adherence to the MD (from 6 to 6.77 points) while evidencing post-intervention improvements from 0.60 to 1.89 points [33,37,44,57,61,62].

The study by Murimi et al., 2018, observed that the most successful interventions, both in primary and secondary schools, shared the use of a multi-component approach, age-appropriate activities, and a good alignment of intervention activities with the stated purposes [73]. On the one hand, all these factors contribute to better contextualizing the intervention to local specific conditions, thus increasing success probabilities. On the other hand, as reported by the present study, if they are not properly described or if the intervention studies are not designed to assess the effectiveness of each component, it is rather difficult to quantify a single component’s influence.

An explanation of the lack of relationships observed by the present study can be partially derived from another factor to consider when evaluating the success of a nutritional education intervention: the role of educators. Research has demonstrated that educators have the greatest impact on student achievements [75,76]. Despite the majority of the research on teachers’ and educators’ effectiveness being focused on technical skills and pedagogical and topic knowledge as the key determinants of effectiveness, a growing body of evidence has shown that increased teacher expertise does not always lead to better student results [77]. For instance, a recent study has shown that teachers’ enthusiasm and interest in the subject are found to influence students’ engagement, which in turn can affect the overall effectiveness of the educational process [76]. None of the studies identified by this research have taken educators’ enthusiasm into consideration. Nonetheless, this emerging aspect evidences the potential benefit for intervention efforts that could come from consideration of educator-level emotional factors that are less understood in terms of their impacts on classroom processes.

All things considered, a standardized approach to NE applied to children and adolescents is still lacking, as well as an understanding of the effectiveness of education tools. To address these inconsistencies, the present work proposes a series of suggestions that foster the use of a systematic approach when NE interventions are designed and implemented:The adoption of a systemic, transdisciplinary, and multi-level approach, both in the design phase and during the implementation of NE programs;The analysis and reporting of the details of school and/or class context, e.g., size, socioeconomic composition, cultures, and values, to allow comparisons with other school settings;An overall and detailed description of the educational program;The description and respective justification for the themes addressed by the programs, even if they are related to the school context;The report of all intervention tools utilized within the program, dividing them into formal and non-formal, and possibly including a description or attachment of all graphic materials as the supplementary information;The report of seminar durations and frequency and interventions;The details of who delivers the lessons and/or who is in charge of each educational activity;The description of the eventual parents’ and teachers’ involvement;An analysis of barriers encountered during the implementation phase, including an evaluation of intervention tools;The use of validated systems for evaluating and quantifying the final effectiveness of interventions, taking into account common variables, using similar measurement criteria.

The adoption of these recommendations when designing and developing an NE program could improve the success of future projects. The identification of successful intervention tools can be used by policymakers and educators to reproduce and upscale local programs to regional and national levels, thus avoiding the disparity in knowledge and the fragmentation of objectives and outcomes [78]. An educational program developed on a framework of useful, detailed information around evidence-based successful factors could yield more accurate results in this area of research while being more cost-effective.

## 5. Strengths and Limits

This study is the first to focus on methodological aspects of nutrition education programs by evaluating the influence of educational tools and other parameters on the effectiveness of the outcome.

Some limitations of this research should be mentioned, mostly related to the process of assessing the influence of intervention tools used in food education programs. The process by which interventions are designed is not based on a standard: each study makes different design choices. There are also different methods of assessing the impact of the instruments chosen, which vary considerably from case to case. Moreover, the tools, duration, and/or frequency of the lessons were not sufficiently described, thus hindering extraction. Also, the limited number of studies identified for quantitative analysis may have reduced statistical power, increasing the likelihood of a type II error. Finally, this study acknowledges no evidence that improvements found in the KIDMED score have any metabolic or weight-reducing benefits.

## 6. Conclusions

Looking at the changes in the eating behaviors of the general population, the promotion of a correct lifestyle for healthy growth since early childhood appears to be a central public health challenge [16,36,50,66,79,80,81,82]. As observed by our study, little is known regarding the most effective approach in the field of nutrition education, and there is still little information on the methodology applied to NE programs in children and adolescents [83]. The results of the impact generated by the programs are often uncertain, the evaluation methods for assessing effectiveness are in most cases not specified, and there is no common method for measuring the influence of educational tools on changes in adherence to healthy eating patterns. These factors are paired with the fact that tools and strategies used in NE intervention studies are generally not justified. When limiting the sample to homogeneous studies focused on the MD, all NE programs showed an improvement in the diet adherence, in some cases even higher than 10%. Of note, these results should be taken with caution, since only 4 of the 11 studies showed an increase of more than a single point on a scale of 0–12, and none increased more than 2 points. Further studies should clarify if the change in eating behaviors is only due to the nutrition education intervention. Moreover, this study found no relationships between the nutritional educational tools and intervention effectiveness, and to improve the scientific approach in this field, it provided an initial contribution to adopting a systematic framework in designing and reporting NE programs.

## Figures and Tables

**Figure 1 nutrients-17-02460-f001:**
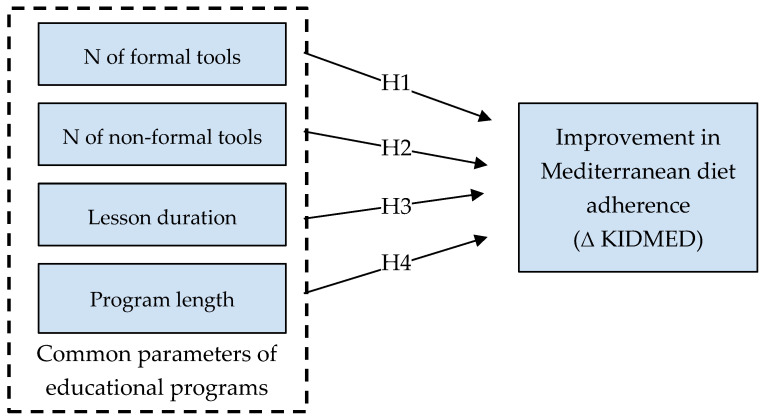
Hypothesis graphical model showing the relationships between variables.

**Figure 2 nutrients-17-02460-f002:**
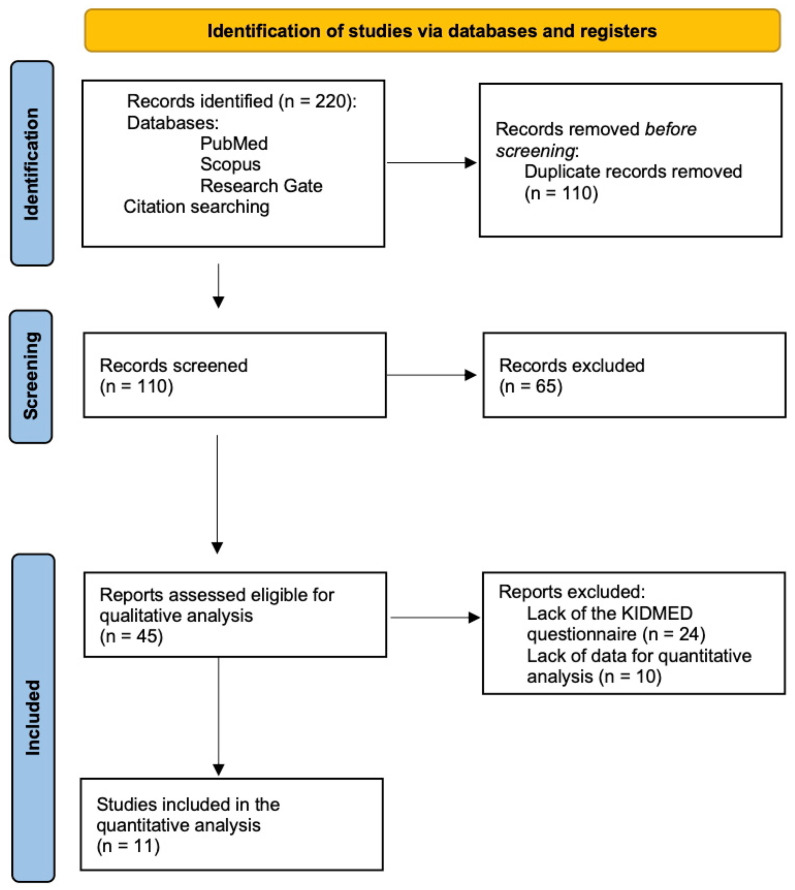
Flow-chart of the study selection process.

**Figure 3 nutrients-17-02460-f003:**
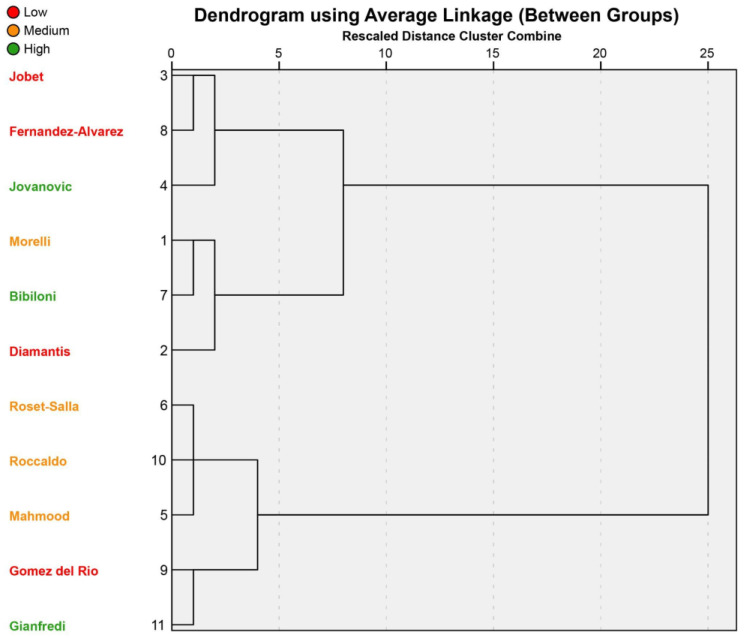
Dendrogram of hierarchical cluster analysis. Improvements in MD adherence are divided into low (≤0.50; red), medium (from 0.51 to 1.00; orange), and high (>1.00; green).

**Table 1 nutrients-17-02460-t001:** KIDMED test to assess the Mediterranean diet quality.

Scoring	
+1	Takes a fruit or fruit juice every day
+1	Has a second fruit every day
+1	Has fresh or cooked vegetables regularly once a day
+1	Has fresh or cooked vegetables more than once a day
+1	Consumes fish regularly (at least 2–3 times per week)
−1	Goes more than once a week to a fast-food (hamburger) restaurant
+1	Likes pulses and eats them more than once a week
+1	Consumes pasta or rice almost every day (5 or more times per week)
+1	Has cereals or grains (bread, etc.) for breakfast
+1	Consumes nuts regularly (at least 2–3 times per week)
+1	Uses olive oil at home
−1	Skips breakfast
+1	Has a dairy product for breakfast (yoghurt, milk, etc.)
−1	Has commercially baked goods or pastries for breakfast
+1	Takes two yoghurts and/or some cheese (40 g) daily
−1	Takes sweets and candy several times every day

Note(s): KIDMED: Mediterranean Diet Quality Index in children and adolescents.

**Table 2 nutrients-17-02460-t002:** Data extraction table of the 11 studies suitable for the quantification of efficacy.

First Author (Year)	Population	Intervention Methods	Lesson Duration/ Program Length	KIDMED at Baseline	Results (Δ KIDMED)	KIDMED Improvement (%)
Bibiloni, M.D.M. (2017) [33]	319 (3–7 y.o.); parents	Cartoons, group discussions, drawings, creation of graphics, crosswords.	15′ (children), 30′ (adults)/3 years	6.51	1.89	15.77
del Río, N.G. (2019) [36]	64 (6–12 y.o.)	Physical activity, video gaming, ClassDojo.	2 h/12 weeks	7.67	0.08	0.77
Diamantis, D. (2023) [19]	12.451 (7–11 y.o.)	Puzzles, crosswords, quizzes, booklets, diaries. Textbook, storytelling, physical activity.	1 h/1 month	5.50	0.20	1.67
Fernández-Álvarez, M. (2020) [52]	319 (14–19 y.o.)	Posters, food wheel, food pyramid, web app, group activities, recipe preparation.	no lessons/6 months	6.34	0.05	0.40
Gianfredi, V. (2024) [57]	18 (5–12 y.o.)	Educational laboratories, cooking classes, workshops, interactive games, theatre laboratories, healthy snacks and gadgets.	2 h/1 year	6.28	1.61	13.43
Jobet, E. (2024) [60]	95 (≥18 m.o.)	Infographics, didactic material.	no lessons/5 months	7.40	0.50	4.17
Kendel Jovanovic, G. (2023) [44]	2709 (10–12 y.o.)	Presentations, brochures, infographics, posters, website, school meetings.	20′/3 weeks + 6–9 weeks of follow-up	6.77	1.18	9.83
Mahmood, M.A. (2022) [58]	70 (8–12 y.o.)	Classes, video, workshops, games, physical activity.	90′/12 weeks	N.R.	0.69	5.75
Morelli, C. (2021) [61]	85 (14–17 y.o.)	Seminars, interactive laboratories, official website, Facebook page.	35′/1 year	6.03	0.93	7.75
Roccaldo, R. (2017) [62]	494 (10–11 y.o.)	Classes.	1 h/6 weeks	6.00	1.00	8.33
Roset-Salla, M. (2016) [37]	206 (1–2 y.o.); 195 parents	Workshops, group games, card games.	90′/6 months	6.50	0.60	5.00

Note(s): abbreviations—N.R.: not reported.

**Table 3 nutrients-17-02460-t003:** Kendall’s tau-b correlation values.

		Formal Intervention Strategies	Non-Formal Intervention Strategies	Class Duration	Intervention Length	KIDMED Improvement
Formal Intervention Strategies	tau coefficient	--				
	Sig. (2-tailed)	.				
Non-formal Intervention Strategies	tau coefficient	0.099	--			
	Sig. (2-tailed)	0.720	.			
Class Duration	tau coefficient	−0.436	0.173	--		
	Sig. (2-tailed)	0.101	0.502	.		
Intervention Length	tau coefficient	−0.046	−0.086	0.000	--	
	Sig. (2-tailed)	0.863	0.737	1.000	.	
KIDMED Improvement	tau coefficient	0.000	0.021	0.114	0.076	--
	Sig. (2-tailed)	1.000	0.934	0.635	0.752	.

Note(s): *: *p* < 0.05, **: *p* < 0.01, ***: *p* < 0.001.

## Data Availability

The original contributions presented in this study are included in the article/Appendix A. Further inquiries can be directed to the corresponding author.

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
