# Peer review of "Evaluation of the Influence of Intervention Tools Used in Nutrition Education Programs: A Mixed Approach"

_nutrients, 2025, doi:10.3390/nu17152460_

Round 1
Reviewer 1 Report
Comments and Suggestions for Authors
This manuscript is a review of Food Education programs which focus on dietary patterns to protect human health and/or the environment.
My biggest concern with this article is a lack of cohesion and clarity regarding the title, purpose, abstract, key variables, and methods.
- The methods search strategy included the keywords AND(children). If you intended to limit the review to children then why did you include 6 studies of adults? If you are not limiting your review to children then the search should be repeated with the keyword “children” removed.
- Your stated purpose lines 71-72 was to determine which FE tools were the most effective in increasing nutrition knowledge. Yet, the effectiveness of interventions was assessed using KIDMED, a tool that measures adherence to the Mediterranean diet, not knowledge.
- What exactly do you mean by food education? How is this different then nutrition education?
- Why does the abstract and introduction include discussion of environmental health? I do not see its relevance to the stated purpose which is to measure the effectiveness of nutrition education interventions. A high-quality diet does not necessarily equal a sustainable diet. In addition, your methods do not address assessment of environmental components.
- Line 19: The term “underage” is confusing. Replace with a term the reader can better understand.
Line 21: The following statement is confusing, “A great heterogeneity was observed in educational program designs and intervention tools, which were usually unmotivated”. Do you mean that the programs/interventions were unmotivating? Unmotivating in what regard? How was this assessed?
Line 24-26: This statement does not align with your introduction, which also includes environmental concerns.
Line 43: Change to: “Obesogenic environments influence eating behaviors, discouraging access to fresh locally grown fruits and vegetables.”
Line 65: What exactly do you mean by “food education”. How is it different than “nutrition education”? Since you are measuring nutrition knowledge, I assume there is a nutrition education component, but what else is included in FE. The FE concept needs more clarity in the introduction.
Line 198: “Within the school context, teachers, as well as parents, were often involved and, sometimes, their involvement consisted in real training.” Clarify “their involvement”. Do you mean teachers and parents involvement, or do you mean parents involvement? In addition, the phrase “consisted in real training” needs clarity. What is “real training”
Line 197-198 and 224-241: What is your reference for the definition and classification of formal vs non-formal tools? In general, formal education has learning objectives and learning is intended and planned. I do not see how a poster can be considered a formal tool. In addition, it would help the reader if the definition and classification were discussed prior to the results (i.e., they would fit better in the methods section).
Line 368: “On the basis of the reported results reported” Delete one of the occurrences of the word “reported”
Comments on the Quality of English Language
Some minor concerns. See main comments.
Author Response
Dear Reviewer,
We thank you for the careful revision of our manuscript. We appreciated all your precise and useful suggestions which we used to revise and modify our paper following the sequence of your comments:
- The methods search strategy included the keywords AND (children). If you intended to limit the review to children then why did you include 6 studies of adults? If you are not limiting your review to children then the search should be repeated with the keyword “children” removed.
We revised the list of included studies and excluded 4 studies that included only adults. Those studies were removed from Supplementary table and Figure 2, as well.
2. Your stated purpose lines 71-72 was to determine which FE tools were the most effective in increasing nutrition knowledge. Yet, the effectiveness of interventions was assessed using KIDMED, a tool that measures adherence to the Mediterranean diet, not knowledge.
Following your comment, we modified the aim of the paper by replacing nutrition knowledge with adherence to healthy diets.
"Thus, the aim of this study is to assess which are the most effective intervention tools to increase adherence to healthy eating patterns through NE programs in the youngest population."
3. What exactly do you mean by food education? How is this different than nutrition education?
Food education refers to the inclusion of other dimensions, such as environmental sustainability of diets, in addition to nutritional aspects. However, following your suggestions, we decided to replace food education (FE) with nutritional education (NE) throughout the manuscript, thus focusing the manuscript exclusively on main purpose of assessing the relationship between the use of educational tools and MD adherence improvement.
4. Why does the abstract and introduction include discussion of environmental health? I do not see its relevance to the stated purpose which is to measure the effectiveness of nutrition education interventions. A high-quality diet does not necessarily equal a sustainable diet. In addition, your methods do not address assessment of environmental components.
We removed any reference to the sustainability dimension of eating patterns either in the abstracts or the introduction.
5. Line 19: The term “underage” is confusing. Replace with a term the reader can better understand.
Thank you for the suggestion, we replaced underage students with schoolchildren
Line 21: The following statement is confusing, “A great heterogeneity was observed in educational program designs and intervention tools, which were usually unmotivated”. Do you mean that the programs/interventions were unmotivating? Unmotivating in what regard? How was this assessed?
We replaced "unmotivated" with "not justified"
Line 24-26: This statement does not align with your introduction, which also includes environmental concerns.
We removed any reference to the sustainability dimension of eating patterns either in the abstracts or the introduction.
Line 43: Change to: “Obesogenic environments influence eating behaviors, discouraging access to fresh locally grown fruits and vegetables.”
We modified line 43 according to your suggestion
Line 65: What exactly do you mean by “food education”. How is it different than “nutrition education”? Since you are measuring nutrition knowledge, I assume there is a nutrition education component, but what else is included in FE. The FE concept needs more clarity in the introduction.
Food education referred to the inclusion of other dimensions, such as environmental sustainability of diets, in addition to nutritional aspects. However, following your suggestions, we decided to replace food education with nutritional education throughout the manuscript, thus focusing the manuscript exclusively on main purpose of assessing the relationship between the use of educational tools and MD adherence improvement.
Line 198: “Within the school context, teachers, as well as parents, were often involved and, sometimes, their involvement consisted in real training.” Clarify “their involvement”. Do you mean teachers and parents involvement, or do you mean parents involvement? In addition, the phrase “consisted in real training” needs clarity. What is “real training”
We have further specified the different trainings in which teachers and parents participated.
“Teachers, as well as parents, were often involved in the educational activities: in some cases, their participation was limited to receive newsletters [13,29] or a one-day nutrition education training [30] in other cases, it consisted of multiple nutrition education sessions [31–35] or other activities such as cooking classes [29,32,36,37].”
Line 197-198 and 224-241: What is your reference for the definition and classification of formal vs non-formal tools? In general, formal education has learning objectives and learning is intended and planned. I do not see how a poster can be considered a formal tool. In addition, it would help the reader if the definition and classification were discussed prior to the results (i.e., they would fit better in the methods section).
We improved the description of formal and non-formal tools, and we moved it from the result to the method section:
“Formal tools refer to educational strategies that are included in a structured and tiered education pathway which maintains a front-of-class approach. This category includes oral presentations, textbooks, webinars, videos, Q&A sessions, group discussion and reflection. Non-formal educational tools, e.g., learning by doing and peer education, focus primarily on intellectual, emotional, social, and behavioral aspects than to cognitive performance, as in formal tools. The learning of the contents happens through activities that stimulate all five senses and artistic expression, as well as by means of games and physical activity, thus overcoming the exclusive and unilateral interaction between student and teacher."
Line 368: “On the basis of the reported results reported” Delete one of the occurrences of the word “reported”
We deleted one of the occurrences.
Kind regards
Reviewer 2 Report
Comments and Suggestions for Authors
Journal Nutrients (ISSN 2072-6643)
Manuscript ID nutrients-3702984
Title Evaluation of the effectiveness of intervention tools used in food education: a mixed approach.
The manuscript addresses a relevant and timely topic—the role of Food Education in mitigating poor dietary patterns and promoting environmental and human health. However, major concerns arise regarding clarity, methodological rigor, and the novelty of the findings.
Major Revision Required
While the topic is timely and potentially impactful, the manuscript currently lacks methodological transparency, conceptual clarity, and strong evidence of novelty. A major revision is needed to improve the scientific rigor, interpretation, and academic communication of this work.
The abstract is verbose and contains ambiguous or imprecise language:
-"Marked by the sign of rising temperatures and environmental disasters" is poetic but lacks academic precision.
-Phrases like “unmotivated” (likely meant to be “unjustified” or “not explained”) and “systematicity” (non-standard term) suggest linguistic and editorial issues.
In the conclusion: The phrase “To the best of our knowledge, this is the first…” appears multiple times, which is unnecessary and makes the conclusion feel redundant.
The authors claim novelty in quantifying the effectiveness of intervention tools in FE. However: They also report no significant relationships were found. Therefore, the impact of the study is weakened. The added value is not clearly demonstrated.
Several prior reviews have explored FE programs in youth (as the authors note), though focusing on outcomes rather than methods. This shift of focus is potentially novel, but not sufficiently developed in the abstract or conclusion.
The methodology is mentioned only briefly, with references to a “qualitative analysis” followed by a “quantitative analysis.”
However, it’s unclear what exact methods were used for data extraction, coding, or statistical analysis.
What were the “common parameters” analyzed? How were these parameters defined or chosen?
The criteria for study inclusion and assessment of bias (e.g., PRISMA, GRADE) are not described—an important omission in any review.
The quantitative analysis appears to be underpowered: "The small number of studies included... may have hindered the observation of clear patterns." If the data pool was too small or heterogeneous, quantification may not have been feasible or appropriate.
Regarding Results Interpretation: the central finding—that there was no significant relationship between intervention strategies and improved adherence—is important, but:
- The authors do not explore why this might be the case.
- Possible explanations (e.g., study quality, cultural variation, lack of standardization in FE programs) are not sufficiently discussed in the abstract or conclusion.
- The conclusion feels tentative: It claims to offer a "general framework" for FE but gives no indication of what that framework entails. This makes the impact statement weak.
- Small number of studies
- Lack of standardization in design and measurement
- Heterogeneity of methods
- Lack of formal methodological framework (e.g., PRISMA, AMSTAR)
- No mention of grey literature or publication bias
- Possible selection bias in which studies were included
- No meta-analytical component or effect size reporting
Recommendations for Improvement
- Clarify the methodology:
- Use established systematic review protocols (e.g., PRISMA).
- Include definitions of key terms like “intervention tools,” “common parameters,” and what constitutes “effectiveness.”
- Refine the language:
- Improve clarity and eliminate redundancy.
- Use precise academic terminology (“unmotivated” → “not justified,” “systematicity” → “systematic framework”).
- Add depth to results interpretation:
- Explore why no significant associations were found.
- Discuss implications for policymakers and educators.
- Substantiate claims of novelty:
- Compare findings with recent reviews.
- Provide a clearer picture of the proposed “framework.”
- Improve the conclusion:
- Summarize key findings more succinctly.
- Avoid repeating claims without new insight.
Author Response
Dear Reviewer,
We thank you for the careful revision of our manuscript. We appreciated all your precise and useful suggestions which we used to revise and modify our paper following the sequence of your comments:
The abstract is verbose and contains ambiguous or imprecise language:
-"Marked by the sign of rising temperatures and environmental disasters" is poetic but lacks academic precision.
Following your suggestion, we reformulated the first sentence of the abstract as follows:
“In a global panorama marked by the progressive rise in obesity, metabolic syndrome, and chronic non-communicable disease prevalence, Nutrition Education (NE) might play a pivotal role in restoring and strengthening the relationship with dietary patterns that protects human health.”
-Phrases like “unmotivated” (likely meant to be “unjustified” or “not explained”) and “systematicity” (non-standard term) suggest linguistic and editorial issues.
We replaced “unmotivated” with “not justified” and “systematicity” with “systematic approach” throughout the manuscript
In the conclusion: The phrase “To the best of our knowledge, this is the first…” appears multiple times, which is unnecessary and makes the conclusion feel redundant.
We eliminated both sentences and reformulated the conclusions.
“Looking at the changes in the eating behaviors of the general population, the promotion of a correct lifestyle for a healthy growth since early childhood appears to be a central public health challenge [15,49,67,68,80–83]. As observed by our study, little is known regarding the most effective approach in the field of nutrition education and there is still little information on the methodology applied to NE programs in children and adolescents [84]. The results of the impact generated by the programs are often uncertain, the evaluation methods for assessing effectiveness are in most cases not specified and there is no common method for measuring the influence of educational tools on changes in adherence to healthy eating patterns. These factors are paired with the fact that tools and strategies used in NE intervention studies are generally not justified. When limiting the sample to homogeneous studies focused on the mediterranean diet, all NE programs showed an improvement in the adherence to MD, in some cases even higher than 10%. However, this study found no relationships between nutritional educational tools and intervention effectiveness and, to improve the scientific approach in this field, it provided an initial contribution to adopt a systematic framework in designing and reporting NE programs.”
The authors claim novelty in quantifying the effectiveness of intervention tools in FE. However: They also report no significant relationships were found. Therefore, the impact of the study is weakened. The added value is not clearly demonstrated.
We rewrote the discussion and conclusion sections with a more balanced approach, and we moved the sentence “This study is the first to focus on methodological aspects of nutrition education programs by evaluating the influence of educational tools and other parameters on the outcome effectiveness.” in the Strengths and limits section.
Several prior reviews have explored FE programs in youth (as the authors note), though focusing on outcomes rather than methods. This shift of focus is potentially novel, but not sufficiently developed in the abstract or conclusion.
We developed this point throughout the discussion section, particularly in the following paragraph:
“Several systematic reviews have previously assessed the effectiveness of nutrition school-based programs. They compare nutrition education with other types of interventions such as changes to food offerings in cafeterias and vending machines, nutrition friendly school initiatives or multicomponent interventions [64–66]. Nutrition education was found to be effective in decreasing body mass index z score while improving the intake of fruit and vegetables [65]. All these reviews were centered on educational programs as a whole, without evaluating tools and strategies encompassed in NE interventions [64–66].”
Metodi
The methodology is mentioned only briefly, with references to a “qualitative analysis” followed by a “quantitative analysis.”
Following your suggestion, we expanded the description of the methodological approach as follows:
“This study is designed as mixed method research that uses a sequential exploratory strategy. It involves a “first phase of qualitative data collection and analysis followed by a second phase of quantitative data collection and analysis that builds on the results of the first qualitative phase” [23]. The mixed approach combines a literature review with a quantitative analysis, that is used to evaluate the influence of NE tools on the intervention improvement in the adherence to healthy diets.”
However, it’s unclear what exact methods were used for data extraction, coding, or statistical analysis.
What were the “common parameters” analyzed? How were these parameters defined or chosen?
To address these suggestions, we reorganized the method section by dividing subsection 2.3 into 2 different subsections: 2.3. Qualitative analysis phase: data collection and extraction and 2.4. Quantitative analysis phase: data collection, extraction, and coding.
“2.3. Qualitative analysis phase: data collection and extraction
The characteristics of the articles considered to meet the inclusion criteria have been collected on an Excel spreadsheet. For each article, the following information have been extracted using a standardized format: First author (year of publication), Objectives, Population, Intervention methods, Lesson duration/Program length, Outcome, Questionnaire, and Results. Intervention tools were classified in two categories: formal and non-formal tools. Formal tools refer to educational strategies that are included in a structured and tiered education pathway which maintains a front-of-class approach. This category includes oral presentations, textbooks, webinars, videos, Q&A sessions, group discussion and reflection (Johnson, M., and Majewska, D. (2022). Formal, non-formal, and informal learning: What are they, and how can we research them? Cambridge University Press & Assessment Research Report). Non-formal educational tools, e.g., learning by doing and peer education, focus primarily on intellectual, emotional, social, and behavioral aspects than to cognitive performance, as in formal tools. The learning of the contents happens through activities that stimulate all five senses and artistic expression, as well as by means of games and physical activity, thus overcoming the exclusive and unilateral interaction between student and teacher (10.1080/14675986.2021.2018171).
2.4. Quantitative analysis phase: data collection, extraction, and coding
For a quantitative analysis of the effectiveness of interventions, it was decided to select homogeneous studies, restricting them to those that used the Mediterranean Diet Quality Index for Children and Adolescents questionnaire (KIDMED). Since the literature did not always give unbiased evidence of the effectiveness of the NE program implemented and, since the methods of measurement were often different, in order to be able to make a comparison among studies, the evaluation focused on those where the measurement of effectiveness had taken place with the KIDMED test. In fact, the KIDMED questionnaire is the most frequently used in the literature on NE in the area of greatest interest for this study: the Mediterranean Sea area. Data were extracted and reported into an Excel table as follows: "First author (year of publication)", “Population”, "Intervention methods", "Lesson duration/Program length", "Results (Δ KIDMED)", and “KIDMED improvement (%)”. To perform the quantitative analysis, the data were further organized into five variables that were identified as common parameters among all selected studies: the number of "Formal intervention strategies” (e.g., frontal teaching tools), the number of “Non-formal intervention strategies” (e.g., active education tools), "Lesson duration” (expressed in minutes), “Program length” (expressed in months), and “KIDMED improvement” (expressed as a percentage).”
The criteria for study inclusion and assessment of bias (e.g., PRISMA, GRADE) are not described—an important omission in any review.
Inclusion and exclusion criteria were improved as follows:
“The eligibility criteria for the inclusion of the publications were as follows:
- peer-reviewed studies involving human populations under the age of 18 years old;
- experimental studies that investigated a nutrition education intervention;
- studies that assessed pre and post intervention the population adherence to healthy diets;
- studies published between January 2014 and March 2024.
No study has been excluded based on the geographical origin to maintain a broad vision on the subject. Studies were excluded in case the nutritional education intervention concerned a specific population or groups of hospital patients, where the improvement was related to a single aspect of nutrition.”
The quantitative analysis appears to be underpowered: "The small number of studies included... may have hindered the observation of clear patterns." If the data pool was too small or heterogeneous, quantification may not have been feasible or appropriate.
We added three references about the use of statistical tests, i.e., kendall’s tau-b correlation and linear regression with small sample sizes and we reformulated the paragraph as follows:
“Therefore, the non-parametric test of Kendall's tau-b was applied to assess correlation between the variables of interest. Due to the limited sample size (N= 11), Kendall's tau-b test was preferred to Spearman’s rho ranking test for its higher robustness to outliers, the narrower confidence intervals it produces and because it performs well with small sample size [27]. A principal component analysis was then used for dimensionality reduction before applying a multiple linear regression to investigate possible relationships between the dependent variable (i.e., the improvement in MD adherence) and the four common parameters extracted from educational programs (independent variables) (Figure 1) [28]”
Regarding Results Interpretation: the central finding—that there was no significant relationship between intervention strategies and improved adherence—is important, but:
- The authors do not explore why this might be the case.
- Possible explanations (e.g., study quality, cultural variation, lack of standardization in FE programs) are not sufficiently discussed in the abstract or conclusion.
We improved the results interpretation by discussing several possible explanations:
"Moreover, the present research did not find any influence on the improvement of diet adherence not only from the intervention length but even from all the other predictor variables, such as educational tools and seminar duration. The correlation analysis observed no significant relation between the variables considered, as well as the cluster analysis revealed a lack of association between program parameters and similar effectiveness in terms of adherence to MD. A possible explanation has been proposed by Fernandez-Alvarez et al., 2020, suggesting that the lack of improvement in the KIDMED score that his study obtained may be attributed to a “ceiling effect”. According to them, the baseline KIDMED score (6.34) was already within an acceptable range and higher than what had been reported in previous studies on comparable populations [53]. However, such adherence is considered as moderate [26], and, therefore, far from the 12 points maximum value. The present study showed results in contrast with that statement: six studies reported a similar baseline adherence to MD (from 6 to 6.77 points) but evidenced post-intervention improvements from 12 to 38 times higher [31,35,43,57,62,63].
The study of Murimi et al., 2018, observed that the most successful interventions, both in in primary and secondary schools, shared the use of a multicomponent approach, age-appropriate activities and a good alignment of intervention activities with the stated purposes [74]. From the one hand, all these factors contribute to better contextualise the intervention to local specific conditions, thus increasing success probabilities. On the other hand, as reported by the present study, if they are not properly described, or if intervention studies are not designed to assess the effectiveness of each component, it is rather difficult to quantify single component’s influence.
An explanation of the lack of relationships observed by the present study can partially derive from another factor to consider when evaluating the success of a nutritional education intervention: the role of educators. Research has demonstrated that educators have the greatest impact on student achievements [76,77]. Despite the majority of the research on teachers’ and educators’ effectiveness being focused on technical skills, pedagogical and topic knowledge, as the key determinants of effectiveness, a growing body of evidence has shown that increased teacher expertise does not always lead to better student results [78]. For instance, a recent study has shown that teachers’ enthusiasm and interest in the subject is found to influence students’ engagement, which in turn can affect the overall effectiveness of the educational process [77]. None of the studies identified by this research have taken educators’ enthusiasm into consideration. Nonetheless, this emerging aspect evidenced the potential benefit for intervention efforts that could come from consideration of educator-level emotional factors that are less understood in terms of their impacts on classroom processes."
- The conclusion feels tentative: It claims to offer a "general framework" for FE but gives no indication of what that framework entails. This makes the impact statement weak.
We reformulated the conclusions as follows:
“6. Conclusions
Looking at the changes in the eating behaviors of the general population, the promotion of a correct lifestyle for a healthy growth since early childhood appears to be a central public health challenge [15,34,53,55,61,72–74]. As observed by our study, little is known regarding the most effective approach in the field of nutrition education and there is still little information on the methodology applied to NE programs in children and adolescents [57]. The results of the impact generated by the programs are often uncertain, the evaluation methods for assessing effectiveness are in most cases not specified and there is no common method for measuring the influence of educational tools on changes in adherence to healthy eating patterns. These factors are paired with the fact that tools and strategies used in NE intervention studies are generally not justified. When limiting the sample to homogeneous studies focused on the mediterranean diet, all NE programs showed an improvement in the adherence to MD, in some cases even higher than 10%. However, this study found no relationships between nutritional educational tools and intervention effectiveness and, to improve the scientific approach in this field, it provided an initial contribution to adopt a systematic framework in designing and reporting NE programs. “
Recommendations for Improvement
- Clarify the methodology:
- Use established systematic review protocols (e.g., PRISMA).
This study is not designed as a systematic review; therefore, it does not follow an established review protocol, such as the PRISMA. For the same reason, a bias assessment was not conducted. However, to give a more systematic approach to the research, we improved the description of inclusion and exclusion criteria following the PICOS model and we adapted PRISMA work diagram to reshape the Figure 2.
- Include definitions of key terms like “intervention tools,” “common parameters,” and what constitutes “effectiveness.”
Following your comment, we defined intervention tools and common parameters in subsections 2.3 and 2.4, respectively.
Regarding what constitutes “effectiveness, we identified a possible misunderstanding between the effectiveness of education programs and the effectiveness of intervention tools. Therefore, we replaced the term “effectiveness” of intervention tools with “influence” both in the title and in the main text.
2. Refine the language:
-
- Improve clarity and eliminate redundancy.
- Use precise academic terminology (“unmotivated” → “not justified,” “systematicity” → “systematic framework”).
We replaced “unmotivated” with “not justified” and “systematicity” with “systematic approach” throughout the manuscript
3. Add depth to results interpretation:
-
- Explore why no significant associations were found.
We explored possible explanations as follows:
“A possible explanation has been proposed by Fernandez-Alvarez et al., 2020, suggesting that the lack of improvement in the KIDMED score that his study obtained may be attributed to a “ceiling effect”. According to them, the baseline KIDMED score (6.34) was already within an acceptable range and higher than what had been reported in previous studies on comparable populations [53]. However, such adherence is considered as moderate [26], and, therefore, far from the 12 points maximum value. The present study showed results in contrast with that statement: six studies reported a similar baseline adherence to MD (from 6 to 6.77 points) but evidenced post-intervention improvements from 12 to 38 times higher [31,35,43,57,62,63].
The study of Murimi et al., 2018, observed that the most successful interventions, both in in primary and secondary schools, shared the use of a multicomponent approach, age-appropriate activities and a good alignment of intervention activities with the stated purposes [74]. From the one hand, all these factors contribute to better contextualise the intervention to local specific conditions, thus increasing success probabilities. On the other hand, as reported by the present study, if they are not properly described, or if intervention studies are not designed to assess the effectiveness of each component, it is rather difficult to quantify single component’s influence.
An explanation of the lack of relationships observed by the present study can partially derive from another factor to consider when evaluating the success of a nutritional education intervention: the role of educators. Research has demonstrated that educators have the greatest impact on student achievements [76,77]. Despite the majority of the research on teachers’ and educators’ effectiveness being focused on technical skills, pedagogical and topic knowledge, as the key determinants of effectiveness, a growing body of evidence has shown that increased teacher expertise does not always lead to better student results [78]. For instance, a recent study has shown that teachers’ enthusiasm and interest in the subject is found to influence students’ engagement, which in turn can affect the overall effectiveness of the educational process [77]. None of the studies identified by this research have taken educators’ enthusiasm into consideration. Nonetheless, this emerging aspect evidenced the potential benefit for intervention efforts that could come from consideration of educator-level emotional factors that are less understood in terms of their impacts on classroom processes.”
- Discuss implications for policymakers and educators.
We discussed implication at the end of the discussion section.
“The adoption of these recommendations when designing and developing a nutrition education intervention could improve the success of future projects. The identification of successful intervention tools can be used by policymakers and educators to reproduce and upscale local programs to regional and national levels, thus avoiding the disparity in knowledge and the fragmentation of objectives and outcomes [79]. An educational program developed on a framework of useful, detailed information around evidence-based successful factors could yielding more accurate results in this area of research while being more cost-effective.”
4. Substantiate claims of novelty:
-
- Compare findings with recent reviews.
We compared our findings with other recent publications:
Several systematic reviews have previously assessed the effectiveness of nutrition school-based programs. They compare nutrition education with other types of interventions such as changes to food offerings in cafeterias and vending machines, nutrition friendly school initiatives or multicomponent interventions [64–66]. Nutrition education was found to be effective in decreasing body mass index z score while improving the intake of fruit and vegetables [65]. All these reviews were centered on educational programs as a whole, without evaluating tools and strategies encompassed in NE interventions [64–66]. On the contrary, the present study focuses on the analysis of single nutrition education tools. The research findings indicate that interventions with a multisectoral approach had positive outcomes in modifying diet-related risk factors for obese children and adolescents [67]. Equally important is the combination of different intervention tools. Multi-component interventions that adopt both lectures/oral presentations and interactive activities/game sessions have a positive impact on children’s nutritional and physical activity behaviours [15]. The most effective strategies often include experiential learning and cross-curricular activities [13]. It appears relevant the use of NE tools and strategies that capture the attention of the students, such as different kinds of group activities and games, especially board games [63]. Game-based learning is well known for the capacity of stimulating creativity, sociability, and cognitive development, and can be an effective tool for adopting healthy behaviours [15,47].
School garden- and interactive game-based nutrition education programs were found to be highly successful in creating active participation in holistic nutrition interventions [7,18]. In this context, parents and teachers seem to exert a great influence over children's eating behaviours. It has been found that the increase in children's food-related knowledge is greater when their guardians' knowledge increases as well [33,52]. Therefore, the adoption of intervention strategies that include families and school personnel can be crucial [42,71]. Besides, since most students have at least one school meal a day, the food offered at school should be part of comprehensive NE programs, following a “whole-school” approach that emphasize the importance of a healthy diet through an holistic interventional framework [22,24,34,54,67]."
- Provide a clearer picture of the proposed “framework.”
Instead of a real framework, we decided to propose a series of suggestions to improve intervention methodology reporting and provide a more systematic approach to this field.
“All things considered, a standardized approach to NE applied to children and adolescent is still lacking, as well as understanding the effectiveness of individual educational tools. To address these inconsistencies, the present work proposes a series of suggestions that foster the use of a systematic approach when NE interventions are designed and implemented:
- the adoption of a systemic, transdisciplinary and multi-level approach, both in the design phase and during the implementation of NE programs;
- the analysis and reporting of the details of school and/or class context, e.g., size, socioeconomic composition, cultures and values, to allow comparisons with other school settings;
- an overall and detailed description of the educational program;
- the description and respective justification for the themes addressed by the programs, even if they are related to the school context;
- the report of all intervention tools utilised within the program dividing them into formal and non-formal, and possibly including a description or the attachment of all graphic materials as supplementary information;
- the report of seminar durations and frequency and interventions;
- the details of who delivers the lessons and/or who is in charge of each educational activity;
- the description of the eventual parent’s and teachers’ involvement;
- an analysis of barriers encountered during the implementation phase, including an evaluation on intervention tools;
- the use of validated systems for evaluating and quantifying the final effectiveness of interventions, taking into account common variables, using similar measurement criteria.”
5. Improve the conclusion:
-
- Summarize key findings more succinctly.
- Avoid repeating claims without new insight.
We improved the conclusion as follows:
“6. Conclusions
Looking at the changes in the eating behaviors of the general population, the promotion of a correct lifestyle for a healthy growth since early childhood appears to be a central public health challenge [15,49,67,68,80–83]. As observed by our study, little is known regarding the most effective approach in the field of nutrition education and there is still little information on the methodology applied to NE programs in children and adolescents [84]. The results of the impact generated by the programs are often uncertain, the evaluation methods for assessing effectiveness are in most cases not specified and there is no common method for measuring the influence of educational tools on changes in adherence to healthy eating patterns. These factors are paired with the fact that tools and strategies used in NE intervention studies are generally not justified. When limiting the sample to homogeneous studies focused on the mediterranean diet, all NE programs showed an improvement in the adherence to MD, in some cases even higher than 10%. However, this study found no relationships between nutritional educational tools and intervention effectiveness and, to improve the scientific approach in this field, it provided an initial contribution to adopt a systematic framework in designing and reporting NE programs."
Kind regards
Reviewer 3 Report
Comments and Suggestions for Authors
The manuscript uses KIDMED methodology to assess previous study regarding food education with focus on Mediterranean diet. It is an interesting topic, while the methodology is relatively novel. I would suggest:
1. It is needed a paragraph to better explain KIDMED. Even more, in the appendix, the questionnaire needs to be added.
2. Taking into account the small sample size (11 studies), it is needed to add a discussion regarding how appropriate to to conduct statistical analysis with 11 data points.
3. The Figure 3 is not needed. In general, small maps are difficult to see and not very informative (the information contained can be added as text).
Regards
Author Response
Dear Reviewer,
We thank you for the careful revision of our manuscript. We appreciated all your precise and useful suggestions which we used to revise and modify our paper following the sequence of your comments:
The manuscript uses KIDMED methodology to assess previous study regarding food education with focus on Mediterranean diet. It is an interesting topic, while the methodology is relatively novel. I would suggest:
- It is needed a paragraph to better explain KIDMED. Even more, in the appendix, the questionnaire needs to be added.
We added the paragraph 2.5 in the method section where the KIDMED questionnaire was described. Moreover, Table 1 reports the 16 questions and the points assigned by each question.
“2.5. KIDMED questionnaire
Studies included in the quantitative analysis assessed the effectiveness of the intervention using the KIDMED questionnaire (Table 1). This questionnaire was developed and validated in 2004 by a group of scholars led by Serra-Majem [26]. As reported by the authors, the KIDMED “index ranges from 0 to 12 and is derived from a 16-item […]. Items indicating negative aspects of the Mediterranean dietary pattern are scored –1, while those reflecting positive adherence are scored +1. Based on the total score, adherence is classified into three categories: (1) >8, indicating optimal adherence to the Mediterranean Diet; (2) 4–7, suggesting a need for dietary improvement; and (3) ≤3, reflecting very poor dietary quality.”[26].
Table 1. KIDMED test to assess the Mediterranean diet quality
Scoring
+1 Takes a fruit or fruit juice every day
+1 Has a second fruit every day
+1 Has fresh or cooked vegetables regularly once a day
+1 Has fresh or cooked vegetables more than once a day
+1 Consumes fish regularly (at least 2–3 times per week)
-1 Goes more than once a week to a fast-food (hamburger) restaurant
+1 Likes pulses and eats them more than once a week
+1 Consumes pasta or rice almost every day (5 or more times per week)
+1 Has cereals or grains (bread, etc.) for breakfast
+1 Consumes nuts regularly (at least 2-3 times per week)
+1 Uses olive oil at home
-1 Skips breakfast
+1 Has a dairy product for breakfast (yoghurt, milk, etc.)
-1 Has commercially baked goods or pastries for breakfast
+1 Takes two yoghurts and/or some cheese (40 g) daily
-1 Takes sweets and candy several times every day
KIDMED: Mediterranean Diet Quality Index in children and adolescents”
- Taking into account the small sample size (11 studies), it is needed to add a discussion regarding how appropriate to to conduct statistical analysis with 11 data points.
We added three references about the use of statistical tests, i.e., kendall’s tau-b correlation and linear regression, with small sample sizes and we reformulated the paragraph as follows:
“Therefore, the non-parametric test of Kendall's tau-b was applied to assess correlation between the variables of interest. Due to the limited sample size (N= 11), Kendall's tau-b test was preferred to Spearman’s rho ranking test for its higher robustness to outliers, the narrower confidence intervals it produces and because it performs well with small sample size [27]. A principal component analysis was then used for dimensionality reduction before applying a multiple linear regression to investigate possible relationships between the dependent variable (i.e., the improvement in MD adherence) and the four common parameters extracted from educational programs (independent variables) (Figure 1) [28].”
- The Figure 3 is not needed. In general, small maps are difficult to see and not very informative (the information contained can be added as text).
Following your suggestion, we removed Figure 3 and added the information in the main text as follows:
“The geographical origin of the studies referred to different continents (Figure 3). A major prevalence was recorded in the Mediterranean area, with nine Spanish, six Italian, two Portuguese, and two Greek studies, respectively. Other studies implemented in Europe included Belgium, Sweden, Poland, Croatia, and Germany. With regard to other continents, six studies were registered in Africa (three of which in Ghana), five in Northern America (US), and two in Australia. Each of the following countries have also been found to have conducted one study: Japan, Lebanon, Indonesia, Tunisia, Cyprus, Malaysia, Iraq, the Philippines, Egypt, China, Taiwan, Brazil, Chile, Iran, and Nepal.”
Kind Regards
Round 2
Reviewer 1 Report
Comments and Suggestions for Authors
The manuscript is much improved. However, edits remain.
Line 11 and 61. No need to capitalize N and E in “nutrition education”.
Line 12 specify what or whose relationship
Line 22 delete the word “the” from measure [the] adherence to…
Line 22 remove the comma after ‘All studies that measured adherence to the MD[,]
Line 32 delete the word “the” from measure [the] modern dietary habits…
Line 37 delete the word “constantly” from “which are [constantly] growing at epidemic levels”.
40-42 Please provide evidence/reference that the move away from local products caused increased portion size. Or reword this statement.
Line 68 the term “youngest population” is also confusing. In your responses you state you replaced “underage students” with “schoolchildren”, but here the term is “youngest population”. I suggest the term “children and adolescents” since your methods included “human populations under the age of 18”.
195 give the number of studies that targeted young students aged between 5 and 12
315-317 to help the reader understand the results better, be more specific with regard to the variables as they are listed in this section. For example, “intervention tools” should be “number of formal tools”, “number of non-formal tools”.
Methods/Results/Discussion section. Eleven observations is a very small number for a correlation study. Was a power analysis conducted? Some of your correlation values are quite high suggesting the results may have been significant if the analysis was adequately powered. Also, it seems very off that the formal intervention strategies/KIDMED improvement correlation = 1 and yet it was not significant. A Kendall's tau-b correlation value of 1 indicates a perfect positive association between two ranked variables. How could a correlation value of 1 not be significant? Please double check and address this issue. Then edit the methods/results/discussion as appropriate.
A systematic review with meta-analysis is usually the preferred method for assessing the combined effects of studies. Please consider this point in your discussion.
Comments on the Quality of English LanguageNumerous English grammar concerns. Some (but not all) include:
The manuscript is much improved. However, edits remain.
Line 11 and 61. No need to capitalize N and E in “nutrition education”.
Line 12 specify what or whose relationship
Line 22 delete the word “the” from measure [the] adherence to…
Line 22 remove the comma after ‘All studies that measured adherence to the MD[,]
Line 32 delete the word “the” from measure [the] modern dietary habits…
Author Response
Dear Reviewer,
We thank you for the careful revision of our manuscript. We appreciated all your precise and useful suggestions which we used to revise and modify our paper following the sequence of your comments:
Line 11 and 61. No need to capitalize N and E in “nutrition education”.
We modified the abstract and introduction section according to your comment.
Line 12 specify what or whose relationship
We reformulated the sentence as follows: “In a global panorama marked by the progressive rise in obesity, metabolic syndrome, and chronic non-communicable disease prevalence, nutrition education (NE) might play a pivotal role in restoring adoption and strengthening adherence to dietary patterns that protect human health.”
Line 22 delete the word “the” from measure [the] adherence to…
We modified the abstract according to your comment.
Line 22 remove the comma after ‘All studies that measured adherence to the MD[,]
We modified the abstract according to your comment.
Line 32 delete the word “the” from measure [the] modern dietary habits…
The word was deleted according to you suggestion
Line 37 delete the word “constantly” from “which are [constantly] growing at epidemic levels”.
The word was deleted according to you suggestion
40-42 Please provide evidence/reference that the move away from local products caused increased portion size. Or reword this statement.
We modified portion size with increased caloric intake and added the following reference for the statement:
“Popkin BM, Ng SW. The nutrition transition to a stage of high obesity and noncommunicable disease prevalence dominated by ultra-processed foods is not inevitable. Obes Rev. 2022;23(1):e13366. doi:10.1111/obr.13366”
Line 68 the term “youngest population” is also confusing. In your responses you state you replaced “underage students” with “schoolchildren”, but here the term is “youngest population”. I suggest the term “children and adolescents” since your methods included “human populations under the age of 18”.
We modified the sentence according to your suggestion:
"Thus, the aim of this study is to assess which are the most effective intervention tools to increase adherence to healthy eating patterns through NE programs in children and adolescents."
195 give the number of studies that targeted young students aged between 5 and 12
We added the number of studies that focused on primary school children: n=28
315-317 to help the reader understand the results better, be more specific with regard to the variables as they are listed in this section. For example, “intervention tools” should be “number of formal tools”, “number of non-formal tools”.
We modified the manuscript according to your suggestion:
“However, Kendal’s correlation and multiple linear regression tests showed a lack of relationship between the number of formal tools, the number of non-formal tolls, seminar and program durations, and the adherence improvement.”
Methods/Results/Discussion section. Eleven observations is a very small number for a correlation study. Was a power analysis conducted? Some of your correlation values are quite high suggesting the results may have been significant if the analysis was adequately powered. Also, it seems very off that the formal intervention strategies/KIDMED improvement correlation = 1 and yet it was not significant. A Kendall's tau-b correlation value of 1 indicates a perfect positive association between two ranked variables. How could a correlation value of 1 not be significant? Please double check and address this issue. Then edit the methods/results/discussion as appropriate.
We thank the reviewer for this comment. Kendall’s tau correlation test can be applied to small sample size, as well as linear regression tests. To address this point, we added three specific references:
“27. Arndt S, Turvey C, Andreasen NC. Correlating and predicting psychiatric symptom ratings: Spearman’s r versus Kendall’s tau correlation. J Psychiatr Res. 1999 Mar 1;33(2):97–104.
- Bujang MA. An elaboration on sample size determination for correlations based on effect sizes and confidence interval width: a guide for researchers. Restor Dent Endod. 2024;49(2):e21. Published 2024 May 2. doi:10.5395/rde.2024.49.e21
- Jenkins DG, Quintana-Ascencio PF. A solution to minimum sample size for regressions. Han G, editor. PLOS ONE. 2020 Feb 21;15(2):e0229345.”
Moreover, we included this aspect as a limitation in the Strength and limit section: “Finally, the limited number of studies identified for quantitative analysis may have reduced statistical power, increasing the likelihood of a type II error.”
Lastly, regarding results: Kendall’s tau values between the four parameters and the KIDMED improvement range from 0.000 to 0.114 (from negligible to weak), while the p-values for these correlations range from 0.635 to 1.000. Results remained unchanged after double checking.
A systematic review with meta-analysis is usually the preferred method for assessing the combined effects of studies. Please consider this point in your discussion.
We thank you the reviewer for this comment. We know the importance of systematic reviews and meta-analyses. However, the aim of this study was not to assess the combined effect of nutrition education intervention studies, but rather to evaluate the possible influence of tools and strategies on the study effectiveness. To do so, we designed this study with a different methodological approach and different statistical tests.
Numerous English grammar concerns. Some (but not all) include:
The manuscript is much improved. However, edits remain.
Line 11 and 61. No need to capitalize N and E in “nutrition education”.
Line 12 specify what or whose relationship
Line 22 delete the word “the” from measure [the] adherence to…
Line 22 remove the comma after ‘All studies that measured adherence to the MD[,]
Line 32 delete the word “the” from measure [the] modern dietary habits…
The manuscript was thoroughly reviewed to improve the English language
Reviewer 2 Report
Comments and Suggestions for Authors
Accept
Author Response
Dear Reviewer,
thank you for all your suggestions.
Kind regards
Reviewer 3 Report
Comments and Suggestions for Authors
Dear authors,
Thank you, I see the effort that you have done to improve the article.
Regards
Author Response
Dear Reviewer,
thank you for the kind comment.
Kind regards